# Molecular basis of cooperativity in pH-triggered supramolecular self-assembly

Yang Li[1], Tian Zhao[1], Chensu Wang[1,2], Zhiqiang Lin[1], Gang Huang[1], Baran D. Sumer[3] & Jinming Gao[1]

Supramolecular self-assembly offers a powerful strategy to produce high-performance, stimuli-responsive nanomaterials. However, lack of molecular understanding of stimulated responses frequently hampers our ability to rationally design nanomaterials with sharp responses. Here we elucidated the molecular pathway of pH-triggered supramolecular self-assembly of a series of ultra-pH sensitive (UPS) block copolymers. Hydrophobic micellization drove divergent proton distribution in either highly protonated unimer or neutral micelle states along the majority of the titration coordinate unlike conventional small molecular or polymeric bases. This all-or-nothing two-state solution is a hallmark of positive cooperativity. Integrated modelling and experimental validation yielded a Hill coefficient of 51 in pH cooperativity for a representative UPS block copolymer, by far the largest reported in the literature. These data suggest hydrophobic micellization and resulting positive cooperativity offer a versatile strategy to convert responsive nanomaterials into binary on/off switchable systems for chemical and biological sensing, as demonstrated in an additional anion sensing model.

[1] Department of Pharmacology, Simmons Comprehensive Cancer Center, University of Texas Southwestern Medical Center, 5323 Harry Hines Blvd., Dallas, Texas 75390, USA. [2] Department of Cell Biology, University of Texas Southwestern Medical Center, 5323 Harry Hines Blvd., Dallas, Texas 75390, USA. [3] Department of Otolaryngology, University of Texas Southwestern Medical Center, 5323 Harry Hines Blvd., Dallas, Texas 75390, USA. Correspondence and requests for materials should be addressed to J.G. (email: jinming.gao@utsouthwestern.edu).

High-performance, stimuli-responsive nanomaterials that can sharply respond to and amplify biological signals are rapidly developed for disease diagnosis and therapy[1–7]. Synthetic approaches employing traditional covalent bond chemistry may be limited in achieving highly complex nanostructures of over $10^6$ Da in molecular weight. In contrast, non-covalent supramolecular self-assembly offers a versatile and modular strategy in generating nanoscale structures and architectures ($10^6$–$10^9$ Da and 10–100 nm) that often display cooperative behaviours, which arises from subtle interplay of a multitude of non-covalent interactions (for example, hydrogen bonding, hydrophobic and electrostatic interactions)[8–10]. The system as a whole behaves quite differently from the sum of individual parts acting in isolation. Despite the great promise, incorporation of self-assembly principles in the design of responsive nanomaterials is challenging due to the lack of fundamental understanding of cooperativity at the molecular level.

pH is an important physiological parameter that plays a critical role in cellular and tissue homeostasis[11]. Dysregulated pH has been recognized as a hallmark of cancer[12]. pH-sensitive nanoparticles have been widely used for tumour imaging, study of endosome/lysosome biology and cancer-targeted drug delivery[13–15]. Conventional small molecular sensors[16], pHLIP peptides[17] or photoelectron transfer nanoprobes[18] offer continuous pH responses, therefore cannot convert the subtle differences in physiological pH into a discrete signal without introducing noise.

In this study, we investigated the molecular basis of cooperativity in pH-triggered supramolecular self-assembly of a series of ultra-pH sensitive (UPS) copolymers. When PR segment reached above a hydrophobic threshold, nanophase separation drove cooperative deprotonation of charged polymers, as demonstrated by divergent proton distribution in either the protonated unimers or neutral micelles. A combination of theoretical modeling and experimental validation confirmed the micellization-induced pH cooperativity and resulting sharp fluorescent transitions. Our studies suggest hydrophobic nanophase separation may serve as a versatile strategy to convert responsive nanomaterials into binary on/off switchable systems for chemical and biological sensing.

## Results

**UPS nanoprobes.** Lysosensor Green is a commonly used small molecular pH sensor. It exhibits 10-fold change in fluorescence intensity over a 2 pH span (Fig. 1a,c). The continuous change of fluorescence intensity hampers its ability to differentiate small pH variations between pathological pH (for example, acidic tumour pH, 6.5–6.9 (ref. 12)) and normal pH (7.4). Recently, we have established a library of UPS nanoprobes with sharp pH transitions that are finely tunable in a broad range of physiological pH (4–8; ref. 19). UPS nanoprobes displayed a sharp on/off pH response (see a specific example of PEO-*b*-PDBA (poly(ethylene oxide)-*b*-poly(2-(dibutylamino) ethyl methacrylate)) in Fig. 1b,c), which was used to amplify tumour micro-environmental signals for the robust detection of a broad range of tumours[20]. The UPS nanoparticles consist of amphiphilic block copolymers, where PEO is poly(ethylene oxide) and PR is hydrophobic block with multiple ionizable tertiary amines (Supplementary Methods; Supplementary Fig. 1). At low pH,

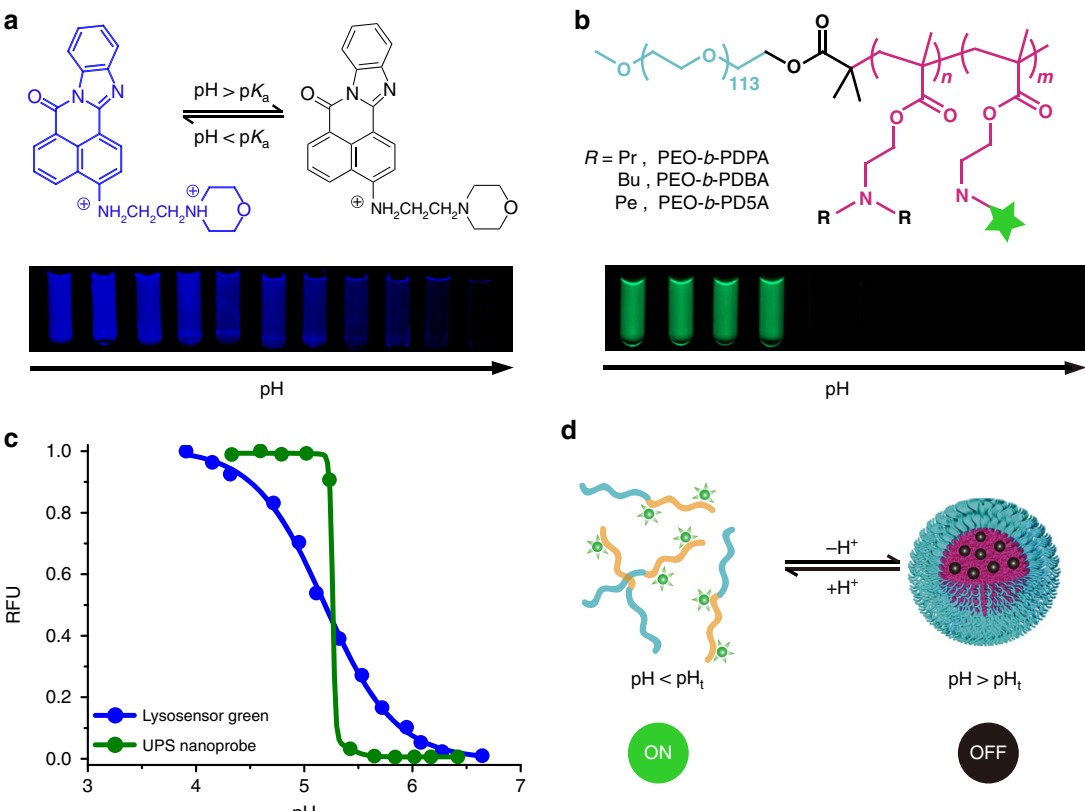

**Figure 1 | Ultra-pH sensitive (UPS) nanoprobes with unique binary on/off response to pH.** (**a**) Structure and fluorescence images of a small molecular pH sensor, Lysosensor Green in aqueous solution at different pH. (**b**) Structure and fluorescence images of a UPS nanoprobe, Rhodamine Green-conjugated PEO-*b*-PDBA block copolymers in aqueous solution at different pH. (**c**) Relative fluorescence intensity as a function of pH for Lysosensor Green and PEO-*b*-PDBA-RhoG nanoprobe. (**d**) Schematic illustration of pH-triggered binary on/off transition of UPS nanoprobes.

micelles dissociate into cationic unimers with protonated ammonium groups (Fig. 1d). When pH increases, neutralized PR segments become hydrophobic and self-assemble into core–shell micelles.

**Micellization is critical for sharp pH transition.** First, we compared the pH responsive behaviours of several UPS copolymers with small molecular and polymeric bases (Fig. 2a; Supplementary Fig. 2). NH$_4$Cl (p$K_a$ = 10.5) and chloroquine (p$K_a$ values = 8.3 and 10.8), commonly used lysosomotropic agents to manipulate the pH of endocytic organelles, showed typical broad pH response in the range of pH 7–11. pH titration of dipropylaminoethanol (DPA, building block of UPS copolymer

poly(ethylene oxide)-b-poly(2-(dipropylamino) ethyl methacrylate) PEO-b-PDPA) showed similar broad pH response (Fig. 2b) as predicted by the Henderson–Hasselbalch equation for monomeric species[21]. pH titrations of several extensively investigated polybases (polyethylenimine (PEI)[22], poly(L-Lysine)[23], chitosan[24] and poly(L-Histidine)[25]) showed different degrees of broad pH response (Supplementary Fig. 2). Among these, PEI had the broadest pH response from from pH 3–11. In contrast, pH titration of three UPS nanoprobes (PEO-b-PDPA, PEO-b-PDBA and PEO-b-PD5A) displayed remarkable pH plateaus within a narrow pH range (Supplementary Fig. 2; Supplementary Table 1), indicating strong buffer effect[26]. For polybases with multiple ionizable sites, the apparent p$K_a$ values were determined as the pH where 50% of all the ionizable amines are protonated[27–29].

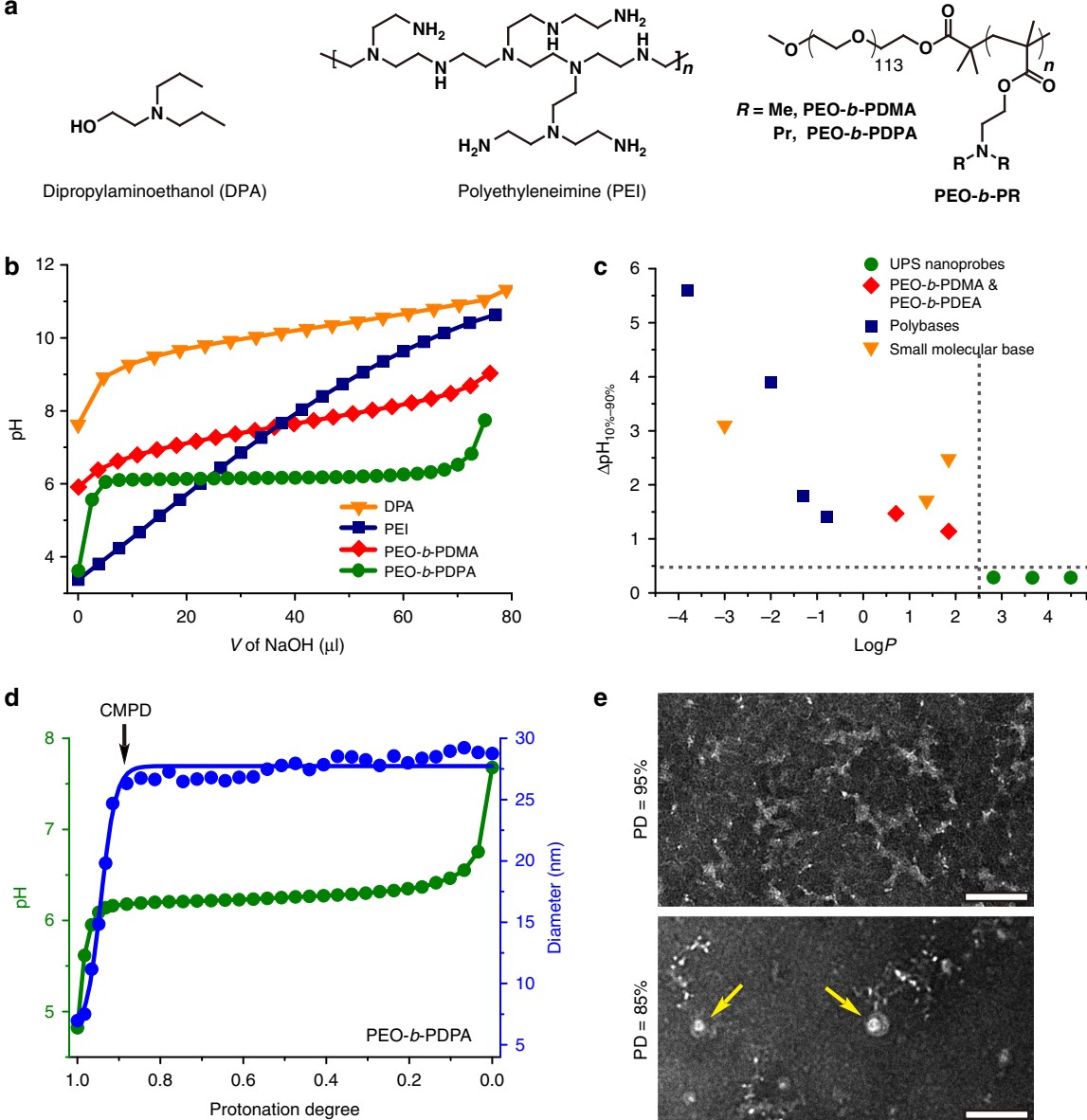

**Figure 2 | Hydrophobic phase separation drives sharp pH response of UPS copolymers.** (**a**) Structures of small molecular base dipropylaminoethanol (DPA), polymeric bases of PEI, PEO-b-PDMA and PEO-b-PDPA. (**b**) pH titration curves of DPA, PEI, PEO-b-PDMA and PEO-b-PDPA. (**c**) Plot of pH transition sharpness (ΔpH$_{10–90\%}$) as a function of octanol–water partition coefficient (Log$P$) of small molecular bases (NH$_4$Cl, Chloroquine and DPA) or repeating unit (neutral monomer) of commonly used polybases (poly(ethyleneimine), polylysine, chitosan, polyhistidine) and PEO-b-PR block copolymers. (**d**) Change of hydrodynamic diameter of UPS nanoprobe PEO-b-PDPA along pH titration coordinate. Significant increase of size indicated the formation of micelles. (**e**) TEM images of PEO-b-PDPA before (protonation degree at 95%) and after (protonation degree at 85%) critical micelle protonation degree (CMPD = 90%). Micelle formation (yellow arrows) was observed when protonation degree was below CMPD. Scale bars, 100 nm.

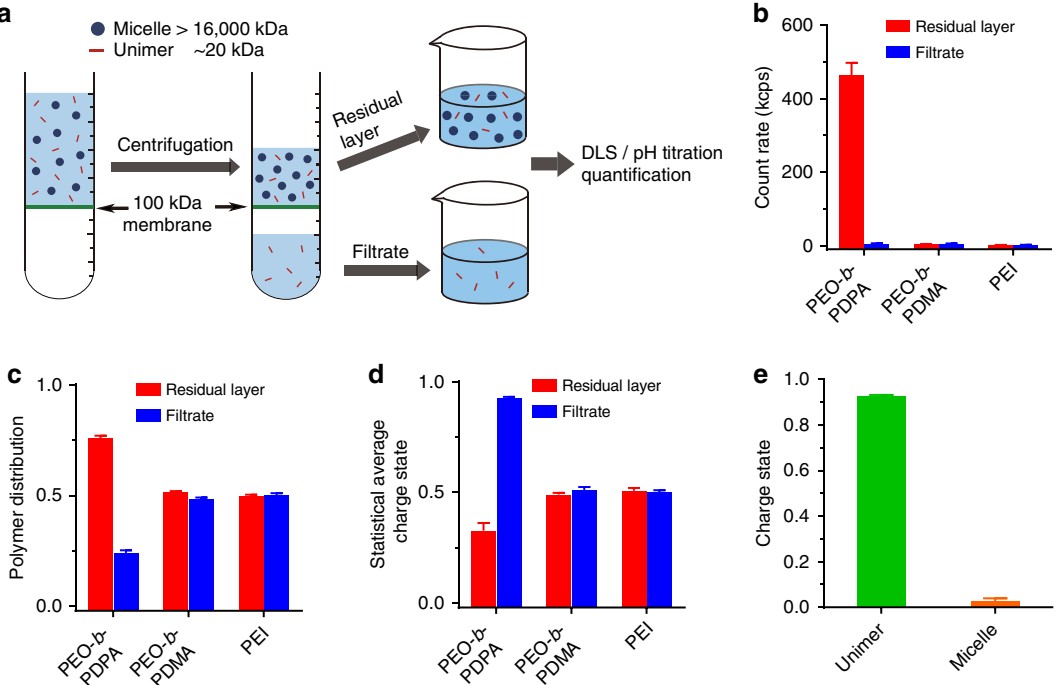

**Figure 3 | Divergent proton distribution between unimer and micelle state of PEO-b-PDPA copolymers.** (**a**) Schematic illustration of dialysis experiments where unimers (22 kD) were separated from micelles (16,000 kD) using a semi-permeable membrane with a molecular weight cutoff of 100 kD. PEO-b-PDMA and PEI were used as negative controls without nanophase separation. Light scattering count rates (**b**), polymer mass distribution (**c**) and charges states (**d**) of different polymers are shown at the protonation degree of 50%. (**e**) Quantification of unimer and micelle charge states of PEO-b-PDPA at protonation degree of 50%. Protons distributed divergently where unimers were highly charged (~90%) and micelles were mostly neutral. The experiments were repeated five times, and data are shown in mean ± s.d.

We calculated the octanol–water partition coefficients (LogP) of small molecular bases or repeating unit (neutral monomers) from polybases and used them as a quantitative measurement of molecular hydrophobicity. We plotted the sharpness of pH transition as defined by $\Delta pH_{10-90\%}$ (the pH span between 10 and 90% ionization of amino groups) as a function of LogP (Fig. 2c). Data indicate that pH sensors with higher molecular hydrophobicity displayed sharper pH responses. Furthermore, a hydrophobic threshold (LogP ~ 2.5) appears to correlate with the ultra-pH response (where we define $\Delta pH_{10-90\%} < 0.5$).

For further investigation, we performed pH titrations of another two PEO-b-PR copolymers with the same methacrylate backbone but less hydrophobic aminoalkyl side chains: methyl and ethyl groups in PEO-b-PDMA (LogP = 1.18) and PEO-b-PDEA (LogP = 1.85), respectively. PEO-b-PDMA displayed a broad pH response from 6.5 to 9.0 (Fig. 2b), where no micelles were formed throughout the titration course. PEO-b-PDEA showed a divergent behaviour (Supplementary Fig. 3), where broad pH response and ultra-pH sensitivity were observed when protonation degree was above and below 60%, respectively. The combination of titration and dynamic light scattering (DLS) results showed that the sharp pH response of PEO-b-PDEA did not start until the formation of micelles, demonstrating correlations between hydrophobic phase separation and ultra-pH sensitivity (Supplementary Fig. 4). Unlike small molecular sensors whose pKa values are mostly controlled by electron withdrawing groups[30,31], the apparent pKa values of UPS copolymers are controlled by the hydrophobicity of PR segment (Supplementary Fig. 5).

DLS method was applied to monitor the formation of micelles along the pH titration coordinate (Fig. 2d; Supplementary Fig. 6). Surprisingly, micelle formation was found as early as 90% of protonation degree and the micelle diameter (~28 nm) remained

relatively constant below 90%. The count rate increased slightly before 92% protonation and increased almost linearly for the remainder of titration. The formation of micelles when protonation degree decreased below 90% was confirmed by transmission electron microscopy (Fig. 2e). Similarly, micelle-induced homoFRET fluorescence quenching did not start until protonation degree reduced to 89% (Supplementary Fig. 7). We define the protonation degree below which the copolymers self-assemble into micelles as the critical micellization protonation degree (CMPD). The CMPD values of PEO-b-PDPA copolymer from different methods showed good consistency.

**Characterization of nanocomplexes along pH titration coordinate.** Intuitively, pH-triggered micellization of PEO-b-PDPA copolymers may undergo two opposite thermodynamic pathways (Supplementary Fig. 8). In the graduate model, below CMPD, majority of PEO-b-PDPA copolymers may first self-assemble into positively charged loose aggregates and upon neutralization, the loose aggregates are gradually deprotonated, shrinking in size and finally turning into neutral, mature micelles. In the divergent model, at any given protonation degree below CMPD, the PEO-b-PDPA copolymers exist in either protonated unimers or neutral, mature micelles with different population distributions. The initial DLS and fluorescence data suggest that pH-triggered self-assembly undergoes the second pathway as indicated by the relatively unchanged nanoparticle size and linearly increased count rate and decreased fluorescence intensity to protonation degree.

To verify the coexistence of protonated unimers and neutral micelles along the pH titration coordinate of PEO-b-PDPA block copolymers, we first applied a dialysis method to separate unimers (22 kD) from micelles (16,000 kD)[20] at different

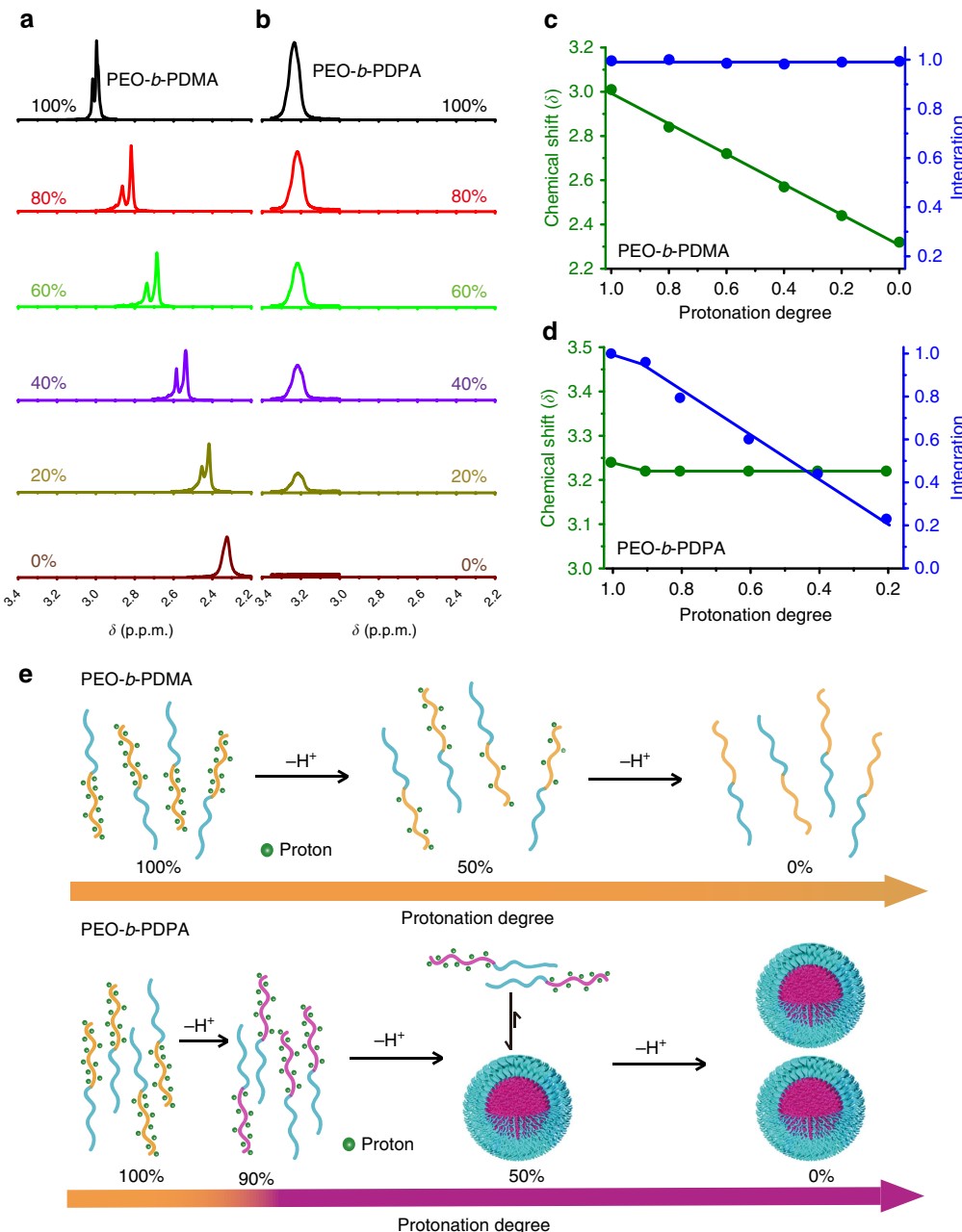

**Figure 4 | Molecular pathway of pH-triggered self-assembly of PEO-b-PDPA copolymers.** (**a,b**) [1]H NMR spectra (in $D_2O$) of methylene protons of PEO-b-PDPA and methyl protons of PEO-b-PDMA adjacent to nitrogen atoms at different protonation degrees, respectively. PEO-b-PDMA was used as negative control without nanophase separation. (**c,d**) Quantification of chemical shift and peak integration of chosen methyl and methylene protons in PEO-b-PDMA and PEO-b-PDPA, respectively. Integrations were calculated using polyethylene oxide (PEO) segments as internal reference. (**e**) Schematic illustration of two distinctive deprotonation pathways. Deprotonation of PEO-b-PDMA ammonium groups was gradual along the entire pH titration course. Deprotonation of PEO-b-PDPA ammonium groups displayed a binary copolymer populations consisting of highly charged unimers in solution and neutral copolymers inside micelles.

protonation degrees (Fig. 3a). Two non-aggregating polybases with broad pH response (PEO-b-PDMA and PEI) were used for comparison. Neither PEO-b-PDMA nor PEI showed changes in size and scattering count rate along its entire titration course (Fig. 3b; Supplementary Fig. 9). Quantitative analysis showed that PEO-b-PDMA and PEI were distributed equally in the residual and filtrate layers (Fig. 3c,d). In contrast, higher amount of PEO-b-PDPA copolymers were retained in the residual layer (Fig. 3c; Supplementary Figs 10 and 11) due to the retention of micelles (16,000 kD, which is above the cutoff molecular weight (100 kD) of the dialysis membrane). pH titration of residual and

filtrate samples showed that PEO-b-PDPA copolymers were highly charged with 90% of tertiary amines protonated in the filtrate sample whereas PEO-b-PDPA copolymers in the micelle state (after subtracting the protonated unimers) in the residual layer were almost neutral (Fig. 3e).

To further confirm the divergent charge state in unimers and micelles, we employed [1]H NMR to study the proton distribution of PEO-b-PDPA along the pH titration path. Data revealed a striking difference in the proton distribution of non-aggregating PEO-b-PDMA versus aggregating PEO-b-PDPA. The PEO-b-PDMA copolymer showed a continuous change of chemical

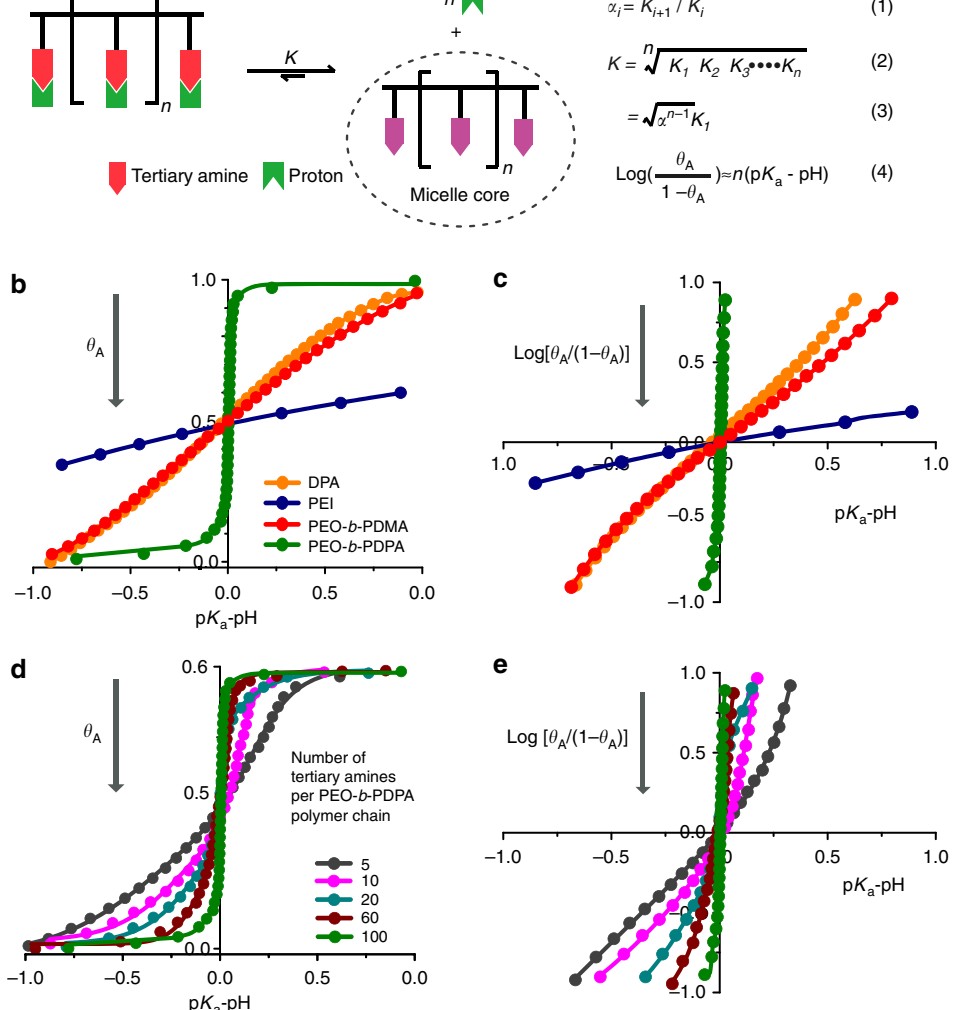

**Figure 5 | Model construction to quantify the pH cooperativity.** (**a**) Micelle-driven deprotonation model with key equations. These equations allow the theoretical-experimental correlation between the microscopic cooperative parameter α and macroscopically measurable $pK_a$. (**b,c**) Cooperativity analysis based on Hill plot of small molecular base DPA, polymeric bases of PEI, PEO-*b*-PDMA and PEO-*b*-PDPA. DPA and PEO-*b*-PDMA showed non-pH cooperativity. PEI displayed negative pH cooperativity and PEO-*b*-PDPA showed strong positive cooperativity. (**d,e**) Cooperativity analysis of PEO-*b*-PDPA copolymers with different number of repeating units in the hydrophobic segment. Increase of hydrophobic chain length led to stronger positive cooperativity and sharper pH response.

shifts of methyl protons along the titration course (Fig. 4a,c; Supplementary Fig. 12). The peak area did not change at different protonation degree (Fig. 4c), suggesting a single phase solution deprotonation process (Fig. 4e). The water soluble small molecular base DPA showed similar results (Supplementary Fig. 13). In contrast, the PEO-*b*-PDPA copolymer displayed divergent proton distribution along the majority of the pH titration coordinate. The unimer peak did not change its chemical shift but only showed linear decrease in integration (Fig. 4b,d; Supplementary Fig. 14; proton peaks in micelle cores were not visible due to fast $T_2$ relaxation), in accordance with the linear increase in scattering count rate and decrease in fluorescence intensity shown previously.

Based on these results, we constructed a molecular pathway of pH-triggered supramolecular self-assembly of PEO-*b*-PDPA copolymer (Fig. 4e). Upon addition of NaOH, the block copolymers are homogeneously deprotonated until reaching CMPD. Below CMPD, further addition of base results in formation of mature micelles where the copolymers inside micelles are neutral. Soluble unimers remain highly protonated at a charge state of CMPD in water. The

unique pH-induced supramolecular self-assembly pathway appeared to pertain microscopic reversibility (Supplementary Fig. 14). It should be noted that the above molecular pathway describes the thermodynamically stable states of polymers at specific protonation degree as measured by steady-state analytical methods (for example, DLS, [1]H NMR). The kinetic process of protonated unimer conversion to neutral micelles may still involve loose aggregates as transient intermediates and needs to be further investigated.

**Model description of pH cooperativity.** The all or nothing proton distribution characteristics[10] suggest a cooperative deprotonation process from fully protonated unimers to the completely neutralized micelles. We hypothesize that rapid deprotonation is driven by hydrophobic phase separation through formation of micelles. We adopted an allosteric model to evaluate the cooperative strength of deprotonation (Fig. 5a; Supplementary Discussion). The progressive neutralization of fully protonated unimers can be characterized by a series of

microscopic $K_i$, which corresponds to the $i$th dissociation constant of polymeric polyatomic acids. The antagonistic or synergistic effect during deprotonation is described by the cooperative parameter $\alpha_i$, which is defined as $K_{i+1}/K_i$ (equation (1) in Fig. 5). The apparent dissociation constant $K$ for the entire deprotonation process is defined by equation (2) (Fig. 5). Assuming identical cooperative parameter $\alpha$ for each ionization step, $K$ can be simplified as equation (3) (Fig. 5). This equation allows a simplified theoretical–experimental correlation between the microscopic cooperative parameter $\alpha$ with macroscopically measurable p$K_a$. Protonation degree ($\theta_A$) is used to describe the extent of protonation of tertiary amines[32]. In practice, the Hill plot is used to evaluate the strength of cooperativity by plotting $\log(\theta_A/(1-\theta_A))$ versus $(pK_a - pH)$ (equation (4) in Fig. 5). The Hill coefficient $n_H$, corresponding to the slope of this plot measured at 50% saturation, is used to quantify the cooperativity strength experimentally[32].

Quantitative analysis revealed that small molecular base DPA had a typical non-cooperative behaviour with a Hill coefficient close to 1 (Fig. 5b,c). In contrast, PEO$_{114}$-$b$-PDPA$_{100}$ displayed exceptionally strong positive cooperativity with $n_H = 51$, compared with 2.3–3.0 for oxygen binding to haemoglobin[33]. Cooperative deprotonation of multiple ammonium groups per polymer chain can dramatically increase the acidification constant as shown in equation (3) (Fig. 5). The p$K_a$ value indeed decreased from 10.1 for monovalent DPA ammonium group to an apparent 6.2 for PEO-$b$-PDPA with 100 ammonium groups per polymer chain—almost 4 order of magnitude increases for the $K_a$ constant. Comparison of PEO-$b$-PDMA copolymer with similar repeating units ($\sim$100) showed a $n_H$ close to 1, indicating non-cooperativity. PEI was found to have strong negative cooperativity with $n_H$ close to 0.3, which is consistent with previously reported neighbouring exclusion effect during protonation of PEI[34].

Next, we investigated the influence of polymer chain length on pH cooperativity. Increasing the number of repeating units in the hydrophobic PR segment significantly sharpened the pH response and increased $n_H$ of PEO-$b$-PDPA copolymers (Fig. 5d,e). In addition, p$K_a$ values of PEO-$b$-PDPA copolymers decreased from 6.66 to 6.26 with the number of repeating unit increased from 5 to 100 (Supplementary Fig. 15; Supplementary Table 2). Changing hydrophobic chain length may provide an additional strategy in fine tuning p$K_a$ and pH transition sharpness.

**Cooperative binding between cationic polymer and anions.** The Hofmeister anion series have been known for their different effects on solubility, stability of secondary and tertiary structures of proteins[35]. We previously reported that chaotropic anions such as perchlorates ($ClO_4^-$) can drive hydrophobic phase separation by pairing with cationic polymers (protonated PEO-$b$-PR copolymers)[36]. In absence of cooperativity, the transition from free anion state to fully bound state will take place over 100-fold ligand concentration as in most monovalent ligand–receptor interactions. The $ClO_4^-$ titration showed 10-fold change in fluorescence intensity in less than threefold in the $ClO_4^-$ concentration span. PEO-$b$-PD5A copolymer with the most hydrophobic side chains displayed strongest cooperativity in $ClO_4^-$ sensing (Supplementary Fig. 16). Moreover, PEO-$b$-PDPA copolymer with higher number of repeating unit (100) showed higher cooperativity than those with shorter chain length (40) (Supplementary Fig. 17). These data showed that hydrophobic nanophase separation can also drive cooperative binding of $ClO_4^-$ anions, indicating its broad potential in the design of binary on/off switchable chemical or biological sensors.

## Discussion

In biological sensing, achieving high sensitivity and specificity of detection remains a great challenge due to the frequently small differences between pathological and normal physiological signals and the difficulty in signal amplification without introducing noise. Unimolecular sensors (for example, Lysosensor) often suffer from low detection accuracy due to lack of signal amplification without noise introduction. Nanomaterials show great promise in addressing the deficiencies of small molecular sensors because of their ability to amplify and switch signals. Self-assembled system is typically more responsive to environmental stimuli due to the lower energy barriers in the formation or dissociation of non-covalent complexes than the making or breaking of covalent bonds. Despite several reported examples[37,38], the design principles for rational conversion of responsive nanomaterials into binary switchable sensors are still lacking. One major obstacle to achieving this goal is the complexity and knowledge gap in the molecular basis of stimuli-triggered responsive behaviours of functional nanomaterials[10].

In this study, we elucidated the molecular pathway of pH-induced supramolecular self-assembly of an ultra-pH sensitive PEO-$b$-PDPA copolymer. Hydrophobic nanophase separation is responsible for the catastrophic deprotonation of charged unimers into neutral copolymers inside polymeric micelles. The divergent proton distribution characteristics of PEO-$b$-PDPA copolymers was not observed in commonly used small molecular and polymeric bases (for example, PEI). The strong pH cooperativity correlated with the significantly decreased p$K_a$ and sharpened pH response compared with unimolecular base (for example, DPA). Combining of theoretical modeling and experimental validation identified key structural parameters such as repeating unit hydrophobicity and polymer chain length that impact p$K_a$ and pH transition sharpness. Cooperativity arising from hydrophobic phase separation was also observed in an anion sensing system, demonstrating the potential universality of this approach.

Cooperativity is a hallmark of self-assembled systems[10,32,39], where the nanosystems often display emergent properties (for example, ultra-pH response) absent from the sum of individual components and can be leveraged to amplify or switch signals[40,41]. Results from this study provide molecular insights to help establish potential principles in applying non-covalent self-assembly chemistry to achieve molecular cooperativity for the development of future nanomaterials-based sensors with sharp response and binary on/off transitions. Conversely, the UPS nanoparticles offer a relatively simple model system to further study molecular cooperativity, which may offer insights in other phase transition systems such as thermosensitive NIPAM[42] and elastin-like polymers[43].

## Methods

**Syntheses of PEO-$b$-PR block copolymers.** PEO-$b$-PR copolymers were synthesized by the atom-transfer radical polymerization method[38]. PEO-$b$-PDPA is used as an example to illustrate the procedure. First, (dipropylamino)ethyl methacrylate (DPA-MA) (1.70 g, 8 mmol), PMDETA (21 µl, 0.1 mmol) and MeO-PEO$_{114}$-Br (0.5 g, 0.1 mmol) were charged into a polymerization tube. Then a mixture of 2-propanol (2 ml) and DMF (2 ml) was added to dissolve the monomer and initiator. After three cycles of freeze-pump-thaw to remove oxygen, CuBr (14 mg, 0.1 mmol) was added into the polymerization tube under nitrogen atmosphere, and the tube was sealed *in vacuo*. The polymerization was carried out at 40 °C for 8 h. After polymerization, the reaction mixture was diluted with 10 ml THF, and passed through a neutral Al$_2$O$_3$ column to remove the Cu catalyst. The THF solvent was removed by rotovap. The residue was dialysed in distilled water and lyophilized to obtain a white powder.

**pH titration.** PEO-$b$-PDPA copolymer (80 mg) was first dissolved in 5 ml 0.1 M HCl and diluted to 2.0 mg ml$^{-1}$ with DI water. NaCl was added to adjust the salt concentration to 150 mM. pH titration was carried out by adding small volumes (1 µl in increments) of 4.0 M NaOH solution under stirring. The pH increase in the range of 2–11 was monitored as a function of total added volume of NaOH. The

fully protonated state and complete deprotonation states (protonation degree equaled 100 and 0%) were determined by the two extreme value points of pH titration curves' 1st derivation. The pH values were measured using a Mettler Toledo pH meter with a microelectrode. Titration of other pH sensitive polymers followed similar procedures using the same amine molar concentration.

**Dialysis.** PEO-b-PDPA copolymers (40 mg) was first dissolved in 2.5 ml 0.1 M HCl and diluted to 2.0 mg ml$^{-1}$ with DI water. PEO-b-PDPA solutions with protonation degree at 0, 25, 50, 75 and 100% were obtained by adding corresponding volumes of 4.0 M NaOH. At each protonation degree, 10 ml polymer solution was centrifuged using ultra-centrifugation tube with a molecular weight cutting-off at 100 kDa to ~5 ml filtrated sample. pH titrations were performed to quantify the amount of polymers and degree of protonation in both residual and filtrate layers. PEO-b-PDMA and PEI were used as control samples and followed similar procedures using the same amine molar concentration. We repeated the experiments five times and data were shown in mean ± s.d.

**NMR analysis.** PEO-b-PDPA copolymer (10 mg) was first dissolved in 0.1 M DCl solution in $D_2O$ and diluted to 2.0 mg ml$^{-1}$ with $D_2O$. NaCl was added to a final concentration of 150 mM. pH titration was performed by adding small volumes (1 μl in increments) of 4.0 M NaOD solution under stirring. Following similar procedures as described above, the volume of NaOD needed to adjust protonation degree to 0, 20, 40, 60, 80 and 100% were calculated based on titration. The NMR spectra were obtained on a Varian 400 MHz $^1H$ NMR Spectrometer at room temperature. NMR analysis of PEO-b-PDMA followed a similar procedure using the same amine molar concentration.

**Data availability.** The authors declare that all other data supporting the findings of this study are available within the article and its Supplementary Information Files, or from the corresponding author upon request.

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

## Acknowledgements

We thank Dr Feng Lin from the NMR Facility at UT Southwestern for help with NMR experiment and all the other members of the Gao lab for thoughtful comments. We thank Dr Daniel Siegwart and Dr Xuewu Zhang for thoughtful discussions. This work is supported by the National Institutes of Health (R01CA192221) and Cancer Prevention and Research Institute of Texas (RP140140).

## Author contributions

Y.L. and J.G. are responsible for all phases of the research. Y.L. synthesized the UPS polymers and performed the pH titration, fluorescence, dialysis, NMR and TEM characterizations. Y.L. also developed the allosteric binding model and performed cooperativity analysis. T.Z., C.W., G.H. and B.D.S assisted with experiment design and data analysis. Z.L. helped with the maestro imaging and dialysis experiments. Y.L. wrote the initial draft. J.G. revised the final draft.

## Additional information

**Competing financial interests:** The authors declare no competing financial interests.

**DOI: 10.1038/ncomms13777**

# Erratum: Molecular basis of cooperativity in pH-triggered supramolecular self-assembly

Yang Li, Tian Zhao, Chensu Wang, Zhiqiang Lin, Gang Huang, Baran D. Sumer & Jinming Gao

*Nature Communications* 7:13214 doi: 10.1038/ncomms13214 (2016); Published 27 Oct 2016; Updated 16 Dec 2016

Figure 5 of this article contained errors introduced during the production process. In Fig. 5c, the minimum value of the *x* axis (left hand side) should be $-1.0$ and not 1.0. Additionally, the maximum value of the *y* axis in Fig. 5d should read 1.0 instead of 0.6. The corrected version of Fig. 5 is shown below as Fig. 1.

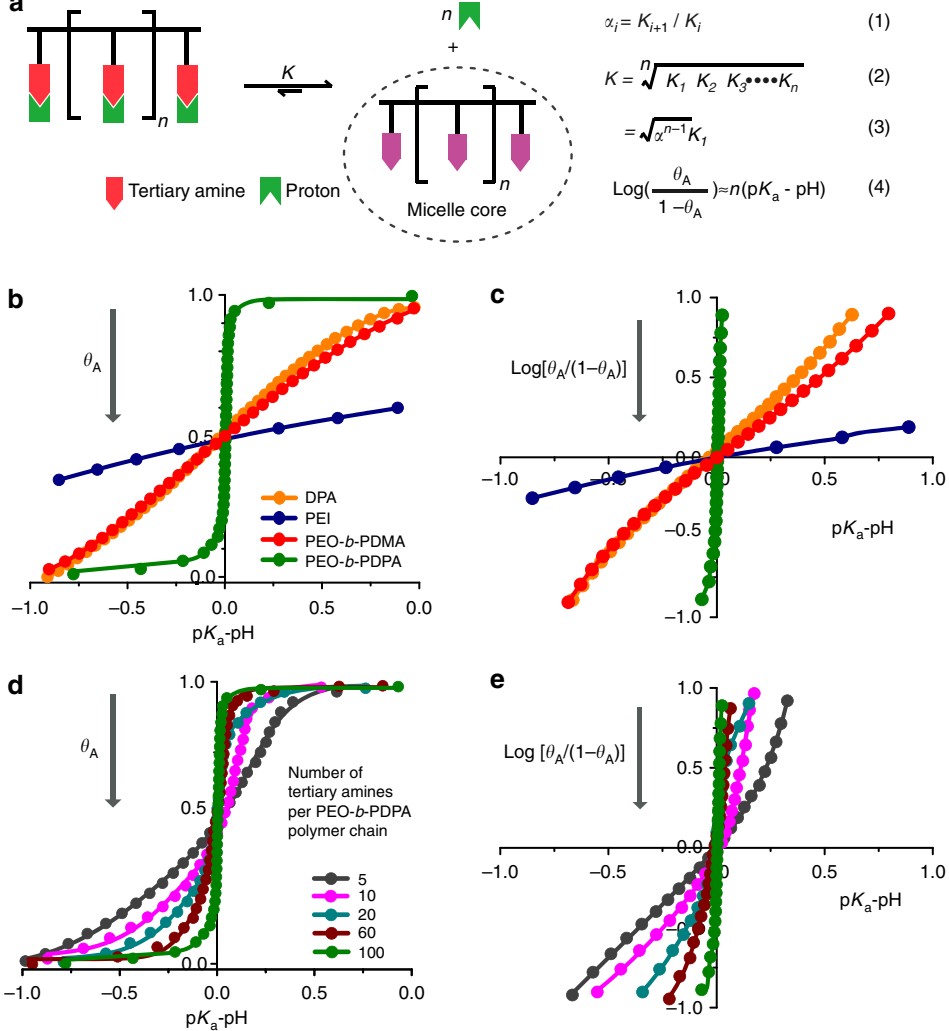

**Figure 1**

