## [Peer Review File · Nature Communications]

Reviewers' comments:

Reviewer #1 (Remarks to the Author):

The authors are commended for a very detailed study on the pH-responsive behavior of their UPS nanoprobe, with a very diligent effort to characterizing this phenomenon and understanding the underlying mechanism. This information adds mechanistic insight into their previously published works on this UPS system.

In general, the work would benefit from a more clear and concise writing style and a more potent and direct abstract and introduction. The figures are clear and the supporting figures contribute to the overall work. The paper would be of interest to others interested in using spontaneous and responsive self-assembly in the creation of probes or delivery agents.

1- Use of the Henderson-Hasselbalch equation is limited in these applications, as pKa for aggregates can not be assumed to be comparable to those of monomeric species. It is recognized that this may be the best approximation the author's can do, but the authors should still comment on the possible confounding results that can be derived from use of this equation for an aggregation species. In fact, much of the data presented by the authors that follows indicates the likelihood that aggregation impacts the expected protonation state of these polymeric sidechains, calling into question the veracity of using an H-H approximation. Additionally, it is unclear how meaningful comparisons to non-aggregating polymeric bases are in this setting.

2- There is a lack of methodological detail throughout for synthesis and use of the polymers. For example, Figure 2c and Supplemental 1e just list use of "different polymers" in this particular assay. This is far too imprecise for a publication of this caliber. At the very least, the authors should provide some basic information on different polymers, including expected monomer percent, fraction modified with dye, and characterization of these polymeric building blocks. Also, the particular dye used is unclear, as was whether a quencher molecule was incorporated, as had been done in their previous papers (ex. JACS 2014). Without this information, a reader is unable to know what may account for a 2-orders of magnitude difference in LogP between "different polymers", for example, as well as it being unclear which polymers are used in many of other experiments shown.

3- The authors should not rule out the possibility of a third pathway that would take into consideration the formation of some loose aggregates that mature over time as electrostatic repulsion is alleviated.

Reviewer #2 (Remarks to the Author):

The manuscript by Li et al. reports an interesting study of the molecular mechanism of the ultra-sensitive pH responsive behavior of block copolymers. Both experimental and theoretical analysis are properly designed and the novelty warrants the publication of this paper in Nature Communications. Thus, I recommend the publication of this work after addressing several minor issues listed below:

(1) In Figure 1a, it could be seen that the probe is still positively charged once the pH of the medium is beyond the pKa of the probe, which would be less beneficial to the aggregation due to electrostatic repulsion. However, quenching of fluorescence clearly indicates the aggregation of probes thereafter. The authors should provide an explanation to this phenomenon.

(2) The detection accuracy will tightly govern the fate of UPS probes during clinical translation. In contrast to in vitro culture, in vivo microenvironment is far more complex. Protein absorption, the high concentration of ions might collectively induce non-specific aggregation for the UPS probes. On this basis, the authors should add some tests to better support the specificity of UPS probes to pH by ruling out these potential interferences.

(3) The aggregation of UPS probes contributes to the quenching of fluorescence, and furthermore the pH-sensing ability. Meanwhile, a higher concentration of UPS monomers is more beneficial to the formation of aggregates once deprotonated. Since inhomogeneous distribution of UPS probes in cell or tissues will likely affect the local concentrations of UPS probes, the author should examine the relationship between the UPS concentration and the pH-sensing performance.

(4) As the authors have shown, the buffering capacity of UPS probes outperforms that of PEI and PEO-b-PDMA. Does this mean that UPS probes might fail to distinguish the pH difference induced by the addition of base in a limited quantity? Also, it is confusing if such strong buffering effect will affect the function of cellular organelles like lysosome since PEI with much less buffering capacity has been proven to exhibit a similar effect.

Response to Reviewers' Comments

We appreciate the thorough and thoughtful comments by the two Reviewers. We carefully considered these comments, and have used them as the basis for this revision. We believe the paper is significantly strengthened as a result of this revision. Here, we provide a point-by-point response to these comments cited in italics.

Reviewer 1:

“The authors are commended for a very detailed study on the pH-responsive behavior of their UPS nanoprobe, with a very diligent effort to characterizing this phenomenon and understanding the underlying mechanism... In general, the work would benefit from a more clear and concise writing style and a more potent and direct abstract and introduction.”

We appreciate the reviewer's positive feedback. Per reviewer's suggestion, we have revised the abstract and introduction to improve the potency and directness of the potential impact of this work. We also revised other sections accordingly to make it more succinct throughout the paper.

“1. Use of the Henderson-Hasselbalch equation is limited in these applications, as pKa for aggregates can not be assumed to be comparable to those of monomeric species. It is recognized that this may be the best approximation the authors can do, but the authors should still comment on the possible confounding results that can be derived from use of this equation for an aggregation species. In fact, much of the data presented by the authors that follows indicates the likelihood that aggregation impacts the expected protonation state of these polymeric side chains, calling into question the veracity of using an H-H approximation.”

We acknowledge the reviewer's concern. We should clarify that we did not intend to use the H-H equation or its approximation to describe the protonation equilibrium of the UPS block copolymers. Instead, we adopted an allosteric model as reported by Anderson and coworkers (*Angew. Chem. Int. Ed.* 2009) to describe the proton binding to UPS polymers with multiple tertiary amine residues. The resulting equation, $\log(\theta_A/(1-\theta_A)) \approx n(pK_a - pH)$ (Equation 4 in the paper), where θ_A and pK_a are protonation degree and apparent acid dissociation constant of combined protonation sites on the UPS copolymer, offers a better description for the protonation equilibrium of UPS copolymers than the H-H equation. Under this model, the apparent pK_a is determined as the pH where 50% of the tertiary amines are protonated ($\theta_A=0.5$). Similar measurement of apparent pK_a 's have been reported in other polymeric bases (e.g., Koper et al in *J. Phy. Chem.*, 1993; *Curr. Opin. Colloid Interface Sci.*, 2006 and Harada et al in *Archives of Biochemistry and Biophysics*, 1979).

To the best of our knowledge, the current paper represents the first report on aggregation-induced all-or-nothing equilibrium process along the pH titration coordinate of a unique class of pH sensitive polymers. Even with the above allosteric model, we agree with the reviewer that there may be other factors that affect the protonation equilibrium. Future work is still warranted to examine how catastrophic phase transitions may require additional modifications to the current allosteric model. We wish to report these studies in future more detailed mechanistic investigations.

“1. Additionally, it is unclear how meaningful comparisons to non-aggregating polymeric bases are in this setting.”

The inclusion of non-aggregating polymeric bases (e.g., PEO-*b*-PDMA) is intended to highlight the importance of hydrophobic phase separation (or aggregation) on the molecular mechanism of pH cooperativity. The non-aggregating polymeric bases like PEO-*b*-PDMA and aggregating polymeric bases like UPS block copolymers have distinctive deprotonation pathways (Fig. 4e). PEO-*b*-PDMA block copolymers were gradually deprotonated along the entire pH titration coordinate, which also applies to other non-aggregating polymeric bases like PEI (Fig S8, S9). In contrast, hydrophobic phase separation from UPS copolymers such as PEO-*b*-PDPA was responsible for the unprecedented pH cooperativity as manifested by the divergent proton distributions.

“2. There is a lack of methodological detail throughout for synthesis and use of the polymers. For example, Figure 2c and Supplemental 1e just list use of "different polymers" in this particular assay. This is far too imprecise for a publication of this caliber. At the very least, the authors should provide some basic information on different polymers, including expected monomer percent, fraction modified with dye, and characterization of these polymeric building blocks. Also, the particular dye used is unclear, as was whether a quencher molecule was incorporated, as had been done in their previous papers (ex. JACS 2014). Without this information, a reader is unable to know what may account for a 2-orders of magnitude difference in LogP between "different polymers", for example, as well as it being unclear which polymers are used in many of other experiments shown.”

We appreciate the helpful suggestions from the reviewer. We have added more detailed descriptions of polymers used in each experiment and figure as highlighted in the revised manuscript. We also emphasized, both in methods section and captions of synthetic scheme, that without specific mention, the hydrophobic chain length of PEO-*b*-PR block copolymers used in the paper was around 80. We added two tables in the supplementary information (Table S1-S2) to provide more detailed physiochemical characterizations of PEO-*b*-PR block copolymers. We also provided a more detailed description of specific dyes used in related experiments. We also

clarified that three fluorophores were conjugated to each PEO-*b*-PR polymer chain and no quencher molecules were incorporated in the current study.

“3. The authors should not rule out the possibility of a third pathway that would take into consideration the formation of some loose aggregates that mature over time as electrostatic repulsion is alleviated.”

We appreciate the inquiry and suggestion from the reviewer. We want to clarify that in our proposed protonation model (i.e., two different pathways in Fig. S7), we refer to the thermodynamically stable states of copolymers at different protonation degree along the pH titration coordinate. Experimentally, these states were characterized by a combination of dynamic light scattering, ultracentrifugation/pH titration, and TEM techniques to quantify the size and charge status of the micelle nanoparticles. All of these methods measure the aggregation states of the copolymers at the equilibrium state. Experimental data demonstrated all-or-nothing divergent states consisting of highly charged unimers or neutral micelles as illustrated in Figure 4e.

The reviewer’s inquiry, which is more related to the kinetic process of how highly protonated unimers convert into mature micelles over time, may occur through two extreme scenarios. One possibility is the charged polymer chains first lose all of their protons in aqueous solution which is followed by the neutral unimers self-assembling into micelles. The other possibility is first the formation of loose aggregates from protonated unimers which then collapse into neutral mature micelles by sudden proton loss. It is also possible that the kinetic pathway follows somewhere between the two extremes. For the micellization process, we agree with the reviewer that loose aggregates should not be ruled out as a possible mechanism for the formation of mature micelles (actually we think it is a more favored pathway). Future work is necessary to elucidate the kinetic pathway and transient intermediates during micelle formation, but may be beyond the scope of this study. We have revised the paper accordingly to clarify that the current model describes the thermodynamically stable states of unimer conversion to micelles.

Reviewer 2:

“1. In Figure 1a, it could be seen that the probe is still positively charged once the pH of the medium is beyond the pKa of the probe, which would be less beneficial to the aggregation due to electrostatic repulsion. However, quenching of fluorescence clearly indicates the aggregation of probes thereafter. The authors should provide an explanation to this phenomenon.”

Figure 1a describes the pH-dependent fluorescence images of Lysosensor Green, a small molecular pH sensor. In this example, deprotonation will not result in dye aggregation, especially

with the unfavorable electrostatic repulsion as pointed out by the reviewer. The pH responsiveness of Lysosensors operates through the photo-induced electron transfer (PeT) mechanism (Urano, *Chem. Rev.*, 2009) where electron transfer from the PeT donor such as tertiary amine to the excited fluorophore decreases the fluorescence intensity. Protonation of amine side chain of Lysosensors relieves the fluorescence quenching. In contrast, the pH-dependent fluorescence response of UPS nanoprobe operates through hydrophobic micellization. In the micelle state, fluorescence intensity is abolished by homo-FRET-induced quenching mechanism. Micelle dissociation into unimers results in dye separation and fluorescence activation. We have reported the detailed fluorescence quenching mechanism in the UPS nanoprobe in a previous publication (*JACS*, 2012).

“2. The detection accuracy will tightly govern the fate of UPS probes during clinical translation. In contrast to in vitro culture, in vivo microenvironment is far more complex. Protein absorption, the high concentration of ions might collectively induce non-specific aggregation for the UPS probes. On this basis, the authors should add some tests to better support the specificity of UPS probes to pH by ruling out these potential interferences.”

We acknowledge the reviewer's concern. We have previously reported the design of two UPS nanoprobe that either target extracellular (UPSe, pKa = 6.9) pH of tumor microenvironment or intracellular acidic organelles of tumor endothelial cells (UPSi, pKa=6.2) (*Nat. Mater.*, 2014). The UPSe can be activated in acidic tumor extracellular fluid (pHe = 6.5–6.8), whereas UPSi can only be activated inside acidic endocytic organelles (for example pH_i = 5.0–6.0). Therefore, the published data from our lab have shown good correlation between *in vivo* tumor imaging data with *in vitro* results. Current work is in progress to evaluate additional biological factors that may impact the *in vivo* imaging efficacy of UPS nanoprobe. However, they are beyond the scope of the current mechanistic study. We wish to publish these data in future reports.

“3. The aggregation of UPS probes contributes to the quenching of fluorescence, and furthermore the pH-sensing ability. Meanwhile, a higher concentration of UPS monomers is more beneficial to the formation of aggregates once deprotonated. Since inhomogeneous distribution of UPS probes in cell or tissues will likely affect the local concentrations of UPS probes, the author should examine the relationship between the UPS concentration and the pH-sensing performance.”

We appreciate the reviewer's concern. Functional nanomaterials for biomedical applications are usually designed to transport therapeutic or diagnostic modalities from the point of administration to the site of action. One potential challenge is to assure that dose dilution in the journey from injection sites to the action sites, such as in blood, will not compromise the

performance of the nanomaterials. In an independent investigation (Figure shown below), we found that the UPS nanoprobe maintained relatively the same transition pH and responsive sharpness from 0.02 mg/ml to 2.0 mg/ml. It is worth noting that the administered concentration of UPS nanoprobe for *in vivo* tumor imaging studies varied from 0.5 to 2.0 mg/ml depending on initial injection dose. The nanoprobe concentration in plasma 24 h after intravenous injection is approximately 0.02 to 0.1 mg/ml (*Nat. Mater.*, 2014). The critical micelle concentration of UPS block copolymers is around 0.001 mg/ml. The polymer concentration in the range of *in vivo* studies (0.02-2.0 mg/ml) does not significantly compromise the performance of the UPS nanoprobe.

“4. As the authors have shown, the buffering capacity of UPS probes outperforms that of PEI and PEO-*b*-PDMA. Does this mean that UPS probes might fail to distinguish the pH difference induced by the addition of base in a limited quantity? Also, it is confusing if such strong buffering effect will affect the function of cellular organelles like lysosome since PEI with much less buffering capacity has been proven to exhibit a similar effect.”

We acknowledge the reviewer’s concern. As shown in Fig. S6, even within the buffer region of the UPS nanoprobe (e.g., where pH is between $pK_a \pm 0.1$), addition of small amount of base can still result in a significant decrease in fluorescence intensity, which can be easily detected by fluorescence imaging method. The buffer effect of UPS nanoprobe inside cellular organelles is dose dependent: at dose $< 100 \mu\text{g/mL}$ in the cell culture medium, we did not observe any significant UPS-induced delay of organelle maturation (*Nature Comm.*, 2015). Under these conditions, the UPS nanoprobe maintains the sharp on/off pH response to sensitively report the reaching of organelle pH to a specific threshold value.

REVIEWERS' COMMENTS:

Reviewer #1 (Remarks to the Author):

The authors are commended for a very detailed study on the pH-responsive behavior of their UPS nanoprobe, with a very diligent effort to characterizing this phenomenon and understanding the underlying mechanism. This information adds mechanistic insight into their previously published works on this UPS system.

The authors have made significant efforts to address the comments from its first review, and the paper has been dramatically improved as a result. The paper and findings therein are of a quality acceptable for publication at this point.

Reviewer #2 (Remarks to the Author):

My concerns have been carefully addressed point by point. Some essential descriptions and discussions have been added into the revised manuscript, make it much better for understanding. I have no further concerns and recommend the publication of this study in this form.